# The Ameliorative Effect and Mechanisms of *Ruditapes philippinarum* Bioactive Peptides on Obesity and Hyperlipidemia Induced by a High-Fat Diet in Mice

**DOI:** 10.3390/nu14235066

**Published:** 2022-11-28

**Authors:** Yahui Song, Qinling Cai, Shanglong Wang, Li Li, Yuanyuan Wang, Shengcan Zou, Xiang Gao, Yuxi Wei

**Affiliations:** 1College of Life Sciences, Qingdao University, Qingdao 266071, China; 2Chenland Nutritionals, Inc., Invine, CA 92614, USA

**Keywords:** *Ruditapes philippinarum*, peptides, obesity, hyperlipidemia, gut microbiota

## Abstract

In this study, bioactive peptides (RBPs) from *Ruditapes philippinarum* were prepared by fermentation with *Bacillus natto* and the effect and mechanisms of RBPs on obesity and hyperlipidemia were explored in mice. We found that RBPs significantly reduced body weight, adipose tissue weight, accumulation of hepatic lipids, and serum levels of total cholesterol (CHO), triglyceride (TG), and low-density lipoprotein (LDL). Mechanistic studies showed that RBPs up-regulated the hepatic expression of genes related to lipolysis, such as hormone-sensitive lipase (HSL), phosphorylated AMP-activated protein kinase (p-AMPK), and peroxisome proliferator-activated receptors α (PPARα), and down-regulated the expression of peroxisome proliferator-activated receptors γ (PPARγ) which is related to lipid synthesis. In addition, RBPs could attenuate obesity and hyperlipidemia by regulating disordered gut microbiota composition, such as increasing the abundance of microflora related to the synthesis of short chain fatty acids (SCFAs) (*Bacteroidetes*, *Prevotellaceas_UCG_001*, *norank_f_Muribaculaceae*, and *Odoribacter*) and controlling those related to intestinal inflammation (reduced abundance of *Deferribacteres* and increased abundance of *Alistipes* and *ASF356*) to exert anti-obesity and lipid-lowering activities. Our findings laid the foundation for the development and utilization of RBPs as a functional food to ameliorate obesity and hyperlipidemia.

## 1. Introduction

In recent years, the human lifestyle has gradually shifted toward the increased dietary intake of high-energy diet and reduced physical activity [1], which have led to an elevated risk of metabolic disorders and related abnormalities, such as obesity and hyperlipidemia [2]. The epidemiological study proposed that hyperlipidemia and obesity will affect nearly 1.12 billion people worldwide in 2030 [3].

The mechanisms of obesity and dyslipidemia are complex with numerous genes involved [4]. To date, researchers have identified many transcription factors as potential therapeutic targets for lipid regulation, including peroxisome proliferator-activated receptors PPARs (PPARα, β, and γ) [5], sterol regulatory element binding protein-1c (SREBP-1c) [6], and others. PPARα participates in the activation of lipolysis and fatty acid β-oxidation, whereas PPARγ and SREBP-1c are involved in the regulation of adipogenesis [7]. Moreover, some enzymes are also pivotal in regulating lipid synthesis and lipid metabolism. AMP-activated protein kinase (AMPK) is an important energy regulatory enzyme. On the one hand, it promotes the expression of PPARα and hormone-sensitive lipase (HSL), which are related to lipolysis. On the other hand, it can directly or indirectly inhibit the expression of genes related to lipid syntheses, such as PPARγ, SREBP-1c, and the downstream fatty acid synthase (FAS) [8,9]. Moreover, HSL is a critical rate-limiting enzyme during lipolysis [10]. By exploring these key transcription factors and enzymes, the regulation of lipid metabolism can be clearly understood.

Recently, changes in gut microbiota composition have been revealed to affect the development of metabolic diseases such as obesity [11]. Previous literature has indicated that the obesity phenotype can be transmitted by transferring gut microbes from obese rodents or humans to recipients [12]. In addition, Org et al. [13] reported that in obese-prone mice, *Akkermansia muciniphila* administration significantly reduced the total cholesterol (CHO) and triglyceride (TG) levels in plasma. Studies have pointed out that the gut microbiota can affect the development of obesity in multiple ways, for example by influencing intestinal permeability, chronic inflammation, energy metabolism, and nutrient absorption [11]. These observations highlight the link between gut microbiota and the development of obesity and hyperlipidemia.

According to the different mechanisms of action, current clinical anti-obesity drugs can be divided into appetite suppressants, fat absorption inhibitors, and energy expenditure stimulators [14]. However, most anti-obesity drugs were eliminated by the market due to various untoward reactions. For example, orlistat can cause digestive system diseases such as flatulence and abdominal pain [15]. Similarly, several drugs for the treatment of hyperlipidemia also have toxic effects [16]. For example, Simvastatin causes muscle pain, and long-term use of the drug can lead to gut microbiota disorders [17]. At present, researchers tend to explore natural substitutes, such as food-derived bioactive peptides to replace synthetic drugs for improving and preventing diseases, including obesity and hyperlipidemia [18]. Anti-obesity and hypolipidemic peptides have been isolated from Skate skin [19], walnut meal [20], silk and silk pupa [21], hazelnut [22], and bovine milk [23,24].

*Ruditapes philippinarum* (*R. philippinarum*) is an important cultured bivalve organism in China, with an annual yield of approximately 3.9 million tons [25]. Studies have shown that the peptides obtained from the hydrolysis of *R. philippinarum* exert multiple bioactivities, such as anticancer, antibacterial, and antihypertensive effects [26]. However, there is no report on the effect of peptides acquired from *R. philippinarum* on obesity or hyperlipidemia. In our previous study, we prepared bioactive peptides (RBPs) by fermenting *R. philippinarum* with *Bacillus natto* and proved the excellent antihypertensive effect of RBPs [26]. This study aimed to explore the ameliorative effects and mechanisms of RBPs in obesity and hyperlipidemia in mice. The finding will provide novel insights into the development of RBPs as a functional food with antiobesity and hypolipidemic effects.

## 2. Materials and Methods

### 2.1. Preparation of RBPs

RBPs were prepared as described in our previously reported method [27]. Briefly, after the fermentation of *R. philippinarum* and sucrose by *Bacillus natto* for 24 h at 45 °C, the supernatant was separated and lyophilized, and the dried powder of RBPs was obtained. The measurement results of each component of the prepared RBPs were shown in Appendix A.

### 2.2. Animal Study

SPF Biotechnology Co., Ltd. (Beijing, China) (SCXK (Jing) 2019-0010) provided healthy 4-week-old Kunming mice (n = 48, males, weighing 24 ± 2 g). All experimental animal use was strictly in accordance with the Animal Care and Use Guide. The mice were fed at the experimental animal platform of the Biomedical Center of Qingdao University and were able to access food and water freely in an SPF room (temperature: 22 °C ± 3 °C, humidity: 65 ± 5%, light/dark cycle: 12 h).

After one week of adaptation, the mice were randomly divided into two groups that were fed with a normal diet (n = 16), and 40% with a high-fat diet (HFD, the composition was shown in Appendix A) (n = 32), respectively. Nine weeks later, mice fed with a normal diet were randomly divided into two groups (n = 8 each): Control (Con) group, gavaged with normal saline; safety evaluation (Saf) group, gavaged with 600 mg/kg RBPs. The mice fed with HFD were randomly divided into four groups (n = 8 each): Model (Mod) group, gavaged with normal saline; Low dose RBPs (LRBP) group, gavaged with 200 mg/kg RBPs; High dose RBPs (HRBP) group, gavaged with 600 mg/kg RBPs; Simvastatin (Sim) group, gavaged with 2.6 mg/kg simvastatin. The mice received corresponding gavage every day at a fixed time for 6 weeks and the treatment dose was referenced to previous publications [20,26].

During the 15 weeks of feeding, the weight of mice was monitored weekly and measured at a fixed time. After the last gavage, the mice were fasted for 12 h and the feces were collected before sacrifice. Subsequently, the blood, heart, liver, spleen, lung, kidney, perirenal fat, and epididymal fat were collected. Each organ and tissue were weighed. The feces, blood, and tissues were stored under appropriate conditions until use.

### 2.3. Biochemical Analysis

The serum levels of CHO, TG, high-density lipoprotein (HDL), low-density lipoprotein (LDL), blood glucose (GLU), glycosylated serum protein (GSP), aspartate aminotransferase (AST), alanine transferase (ALT), total bile acid (TBA), and total bilirubin (TBIL) were determined by an automatic biochemical analyzer (Rayto Life and Analytical Sciences Co., Ltd., Shenzhen, China).

### 2.4. Histopathological Analysis

The different fixed tissues were washed in a disposable dehydration box for several hours, dehydrated with gradient alcohol, and fixed with paraffin wax. The fixed tissue was sliced and stained with hematoxylin and eosin.

For oil red O staining, the fixed tissue was first dehydrated by 20% and 30% sucrose solution, then embedded with OCT embedding agent, pre-cooled on a frozen slicer, and sectioned into slices, which were successively placed in distilled water, 60% isopropanol aqueous solution, oil red O working solution, 60% isopropanol aqueous solution, distilled water, and hematoxylin, and finally sealed with glycerol gelatin.

### 2.5. Immunohistochemical Analysis

The liver sections after dewaxing, antigen retrieval, and serum blocking were incubated with the antibodies of target genes, treated with peroxidase-conjugated streptavidin (Nichirei Biosciences Co., Tokyo, Japan), and stained with DBA reagent and hematoxylin. Photomicrographs were taken with a microscope.

### 2.6. Gut Microbial Sequencing

The processing and sequencing of gut microbial samples were supported by Shanghai Majorbio Bio-Pharm Technology Co., Ltd. (Shanghai, China). The results were analyzed on the Illumina MiSeq platform (Illumina, San Diego, CA, USA).

### 2.7. Statistical Analysis

The results were expressed in mean ± standard deviation. Unless otherwise specified, all data were analyzed by one-way analysis of variance (ANOVA) followed by Tukey’s post hoc test. Statistical significance was recognized for *p*-values of <0.05.

## 3. Results

### 3.1. Effects of RBPs on Growth Parameters of HFD-Fed Mice

As shown in Table 1, before the RBPs intervention was applied, the weight of HFD-fed mice was higher than that of mice fed with a normal diet (*p* < 0.05). After the intervention, the final body weight of the HRBP group and Sim group were markedly reduced compared with the Mod group (*p* < 0.05 for both). In addition, the heart index and liver index of mice were significantly increased after long-term HFD treatment (both *p* < 0.05) and were notably improved by high-dose RBPs intervention (both *p* < 0.05). However, Simvastatin only had an ameliorative effect on the liver index (*p* < 0.05, Table 1). Moreover, the results of the Saf group showed that gavage of high-dose RBPs had no adverse effect on the growth parameters of normal diet-fed mice, which indicated that RBPs had no harmful effects on normal mice based on the current data.

### 3.2. RBPs Treatment Ameliorated Hyperlipidemia in HFD-Fed Mice

At the end of the experiment, the serum CHO, TG, HDL, and LDL levels in the Mod group were higher than those in the Con group (*p* < 0.05) (Figure 1A–D). RBPs treatment notably ameliorated the elevated serum CHO, HDL, and LDL levels (*p* < 0.05). In the Sim group, hyperlipidemia was also improved. We also determined the serum levels of GLU and GSP in mice from each group, as shown in Figure 1E,F. No differences in the serum levels of GSP and GLU were observed among the six groups.

### 3.3. RBPs Reduced Obesity in HFD-Fed Mice

White adipose tissue around the epididymis and kidney was isolated from the mice. As shown in Figure 2C,D, the weights of the epididymal and perirenal adipose tissue in the Mod group were 2.15 times (*p* < 0.05) and 1.67 times (*p* < 0.05) higher than those in the Con group, respectively. Low- and high-dose RBPs treatments significantly decreased the weight of epididymal (55.51% and 71.32%) and perirenal adipose tissue (63.16% and 70.53%), as compared to the Mod group. Moreover, the HE staining of epididymal adipose tissue from the mice revealed that the RBPs notably reduced adipocyte size. After treatment with RBPs, adipocyte size in the HRBP group was reduced by 27.45% (*p* < 0.05) as compared to the Mod group.

### 3.4. Intervention of RBPs Ameliorated Liver Injury in HFD-Fed Mice

We performed histopathological analysis on the liver of mice and measured the level of enzymes related to liver injury in the serum. Liver HE staining (Figure 3A) displayed that compared to the Con group, the liver of mice in the Mod group showed noticeable hepatic steatosis, which was reflected in the disordered arrangement of hepatocytes, and a certain number of lipid droplets were observed. After treatment with RBPs, the number of lipid droplets in the Mod group decreased, indicating reduced hepatic steatosis. Consistent with these results, the result of oil red O staining indicated that HFD caused severe fatty accumulation in the liver and the proportion of lipid droplets (red areas) in high-dose RBPs-treated groups was reduced by 74.12% (Figure 3C). Moreover, elevated serum levels of ALT, TBA, and TBIL were found in the Mod group (*p* < 0.05), reflecting the injury of the liver. The intervention of RBPs and Simvastatin remarkably suppressed the elevation of ALT, TBA, and TBIL in HFD-fed mice (*p* < 0.05 for all).

### 3.5. Effect of RBPs on the Expression of Genes Related to Lipid Metabolism in Liver

To explore the mechanisms of lipid regulation and liver protective function of RBPs, the expression of p-AMPK, SREBP-1c, HSL, FAS, PPARα, and PPARγ in the liver was measured by immunohistochemistry and the average optical density was calculated (Figure 4A–C). Compared with the Con group, the expression of p-AMPK, HSL, and PPARα in the Mod group decreased by 69.86%, 37.50%, and 71.87% (*p* < 0.05), whereas the expression of FAS and PPARγ increased by 37.48% and 45.94% (both *p* < 0.05), respectively. The expression of p-AMPK, HSL, and PPARα increased by 130.48%, 69.03%, and 137.36% in the LRBP group (*p* < 0.05 for all) compared with the Mod group. The high-dose RBPs intervention increased the expression of p-AMPK, HSL, and PPARα by 159%, 165%, and 300% (*p* < 0.05), respectively, and decreased that of PPARγ by 35% (*p* < 0.05). Simvastatin also elevated the levels of p-AMPK, FAS, HSL, and PPARα, and decreased the level of PPARγ (*p* < 0.05). No change in the expression of SREBP-1c was found among the six groups.

### 3.6. RBPs Reduced the Imbalance of Gut Microbiota in HFD-Fed Mice

We collected 1,426,295 qualified sequences in the 16S rRNA from the fecal samples to examine the changes in gut microbiota composition of mice in each group. The results showed that the average length was 420 bp and 99% of the qualified sequences were between 401 and 440 bp. The coverage index (Table 2) showed that the sequencing results covered more than 99.7% of the sequences, which indicated that our results were reflective of the real-world distribution of microbial flora in the samples.

The results of alpha diversity analysis (Table 2) showed that the species abundance of the gut microbiota in the Mod group was significantly decreased compared with mice fed a normal diet, as reflected by the significant decrease in the Chao index and Ace index (both *p* < 0.05). The abundance of gut microbiota increased significantly after the high-dose RBPs intervention (*p* < 0.01), and the ameliorative effect was better than Simvastatin (*p* > 0.05). No change was found in the diversity of gut microbiota of mice, as there was no difference in the Shannon index and Simpson index among groups.

Three different analysis methods based on the abundance of OTUs were used for further evaluation of the changes in gut microbial composition among the different groups, including principal component analysis (PCA), principal co-ordinates analysis (PCoA), and non-metric multidimensional scaling analysis (NMDS) (Figure 5C–E). All the results displayed that the gut microbial composition in the Mod group was completely distinct from that of the Con group, indicating the great differences in microbial composition between the two groups. Slight differences were noticed between the HRBP group, Sim group, and Con group, indicating that the intervention of Simvastatin and RBPs may modify the gut microbiota composition of HFD-fed mice.

The Venn diagram also presented similar results (Figure 5A). The four groups have 388 shared OTUs. The Mod, Sim, and HRBP groups, respectively, shared 13, 17, and 35 identical OTUs with the Con group (Figure 5B). The RBPs intervention increased the number of OTUs shared by the Mod group and Con group, which confirmed that RBPs increased the similarity of the gut microbiota composition between HFD mice and normal mice.

Furthermore, the bacterial abundances at different taxon levels were determined, and 13 known phyla were identified (Appendix A). The relative abundances of the five most abundant microbiota at the phylum level in four groups were presented in Figure 5G. The results showed that HFD increased the proportion of *Firmicutes* in the gut microbiota by 37.57% and decreased *Bacteroidetes* proportion by 48.06%. The proportion of *Firmicutes* decreased by 27.43% and 25.47% and the proportion of *Bacteroidetes* increased by 33.11% and 36.15% after intervention with Simvastatin and high-dose RBPs, respectively. It should be noted that the abundance of *Campilobacterota* and *Deferribactota* was higher in the feces of mice in the Mod group than that in the Con group (*p* < 0.05). After RBPs intervention, the abundance of *Campilobacterota* and *Deferribactota* decreased by 56.17% and 90%, respectively, whereas Simvastatin had no effect on the abundance of these florae.

The five most abundant genera were *norank_f_Muribaculaceae*, *Lactobacillus*, *Lachnospiraceae_NK4A136_group*, *unclassified_f_Lachnospiraceae*, and *Dubosiella* (Figure 5F). Student’s *t*-test was used to compare the differences between the top 15 bacteria at the genus level among the different groups (Figure 6). In the Mod group, the abundances of *norank_f_Muribaculaceae*, *Muribaculum*, *Lachnociostridium*, *Escherichia-Shigella*, and *norank_f_norank_o_Clostridia_vadinBB60_group* were markedly lower than those in the Con group (Figure 6A). RBPs and Simvastatin treatment notably increased the abundance of *norank_f_Muribaculaceae* and *Muribaculum* in mice fed with HFD (*p* < 0.05) (Figure 6B,C). In addition, compared to the Mod group, RBPs also increased the abundance of *Alistipes*, *norank_f_UCG_010*, *Odoribacter*, and *ASF356* (*p* < 0.05) (Figure 6B).

## 4. Discussion

This study first proved that peptides prepared from *R. philippinarum* ameliorated obesity and dyslipidemia induced by HFD in mice. Herein, we reported that the intervention of RBPs reduced the body weight and white adipose tissue weight of HFD-fed mice, and ameliorated hyperlipidemia. The mechanistic study showed that RBPs increased the liver expression of genes related to lipolysis, including p-AMPK, HSL, and PPARα, and decreased that of PPARγ, which was related to lipid synthesis. In addition, RBPs may have anti-obesity and antihyperlipidemic effects by improving the disorder of gut microbiota in HFD-fed mice.

Long-term intake of an HFD will lead to obesity and hyperlipidemia [28]. Thus, an HFD is commonly used to establish models of obesity and hyperlipidemia in animals [29]. Our results were consistent with reports that HFD can lead to excessive weight gain by increasing the volume of adipose tissue [30]. The RBPs intervention slowed weight gain in HFD-fed mice. The histopathological analysis confirmed that RBPs reduced the adipose tissue weight and adipocyte size in mice, indicating the anti-obesity effect of RBPs. In addition, the excessive intake of HFD disrupts the balance of lipid metabolism, and leads to the accumulation of cholesterol and triglycerides, causing dyslipidemia [31]. Dyslipidemia is characterized by the increase in TG, CHO, and LDL and the decrease of HDL in the serum [32]. Epidemiological studies have shown that dyslipidemia was correlated with an elevated risk of cardiovascular disease [33]. For instance, LDL and HDL levels can affect the development of atherosclerosis [34]. Our results showed that RBPs could improve dyslipidemia by reducing the serum levels of CHO, TG, and LDL. This result was similar to that of Hu et al. [19], who described the effect of peptides derived from the skin of Skate (*Raja kenojei*) on HFD-fed mice. Notably, there was an abnormal increase of HDL in the HFD-fed mice, which was similar to that observed by Yeo et al. [35]. Herein, RBPs ameliorated the abnormal elevation of HDL levels in serum. Studies have shown that there is a U-shaped relationship between HDL and cardiovascular disease, and abnormally elevated HDL levels are related to increased cardiovascular risk [36]. Moreover, HFD can induce pathological accumulation of fat in the liver and lead to hepatic steatosis associated with obesity and hypertriglyceridemia [37]. We found that RBPs improved hepatic steatosis, as shown by the alterations in the liver morphology (reduced liver weight), histology (reduced lipid droplet accumulation), and serum biochemical tests (reduced content of ALT, TBA, and TBIL).

The imbalance between lipid synthesis and catabolism induces disorders of lipid metabolism, resulting in obesity, hyperlipidemia, and other diseases [19]. Lipogenesis is mainly regulated by PPARγ and SREBP-1c, which control adipogenesis by affecting the expression of downstream genes, such as FAS, fat acid binding protein 2 (aP2), and acetyl-CoA carboxylase (ACC) [38,39]. Another strategy to improve lipid accumulation is the acceleration of lipolysis in the body. During lipolysis, the expression of PPARα will be highly enhanced and directly regulate the expression of proteins associated with β-oxidation and cholesterol breakdown, such as lipoprotein lipase (LPL), acyl-CoA oxidase (ACO) and carnitine palmitoyltransferase-1 (CPT-1) [40,41]. HSL is a key enzyme that regulates lipid catabolism [42] and is considered a potential therapeutic target for obesity and hypertriglyceridemia [43]. The substrates of HSL include triglycerides, diglycerides, and monoglycerides, as well as cholesterol and steroids [44]. AMPK has a primary role in metabolic regulation and is a key target in obesity prevention [19]. In contrast, AMPK phosphorylation can control the lipogenesis pathway by regulating SREBP-1c to inhibit FAS activity directly, thereby reducing the TG level. Moreover, the activation of AMPK can up-regulate the transcriptional activity of PPARα and lead to the activation of the lipolysis pathways [45]. In this study, RBPs significantly enhanced the hepatic expression of p-AMPK, PPARα, HSL, and FAS, and decreased that of PPARγ in mice, indicating that RBPs promote the process of lipolysis and inhibit the lipogenesis pathway.

The gut microbiome has critical roles in protecting human health and preventing metabolic diseases [46]. Studies showed that the body weight of germ-free recipient mice is affected by the microbiota of different mice (obese or lean mice), confirming that the gut ecosystem plays a critical role in weight management [47]. Gut microbiota was also shown to regulate blood lipids by participating in the bile acid cycle and lipid metabolism [47]. Shi et al. [17] reported that the peptides isolated from Huangjiu exert lipid-lowering activity by regulating the gut microbiota. Recently, Li et al. found that a nine-amino acid peptide from human α-defensin 5 (HD-5) could prevent obesity and reduce blood lipids by modulating the gut microbiota in both rats and macaques [48,49]. These studies suggest the potential for RBPs to improve obesity and hyperlipidemia by regulating the gut microbiota. Decreases in serum TC, TG, and HDL have been reported to be related to increased gut microbiota richness [46], which is consistent with our observations.

Obesity is associated with specific changes at the phylum level of the gut microbiota [12]. In this study, the intervention of RBPs reduced the abundance of the *Firmicutes*, *Actinobacteriota*, *Campilobacterota*, and *Deferribacterota*, and increased that of *Bacteroidetes*. The sum of *Firmicutes* and *Bacteroidetes* accounts for more than 90% of the gut microbiota of humans and mice, and they are the two most important phyla [50]. Studies have reported that some unabsorbed polysaccharides in the diet can be metabolized by *Bacteroidetes* to produce short chain fatty acids (SCFAs) and therefore promote the resolution of obesity and hyperlipidemia [51,52]. SCFAs can promote the secretion of leptin, which can enhance lipid export and regulate insulin sensitivity, and thus regulate disorders of lipid metabolism [53]. In addition, GPR43 is a receptor of SCFAs that can promote lipolysis and inhibit adipogenesis in adipose tissue after being activated by SCFAs [53]. In a previous study, it was pointed out that SCFAs produced by the gut microflora can inhibit the activity of hepatic lipogenic enzymes and regulate cholesterol distribution to reduce levels of TG and TC in serum [54,55]. Moreover, a previous study reported that SCFAs can activate AMPK, which may be another way to regulate blood lipids [45]. The increased fecal abundance of *Firmicutes* may promote the storage of fat in the host [56]. It is suggested that obese animals have higher fecal levels of *Firmicutes* and lower *Bacteroidetes* than lean controls [57], which is similar to our results. *Deferribacterota* and *Campilobacterota* are reported to be related to inflammation [58], which is closely correlated with obesity and hyperlipidemia [59]. *Campilobacterota* is associated with gastroenteritis [60], whereas the increased abundance of *Deferribacteres* is accompanied by the aggravation of intestinal inflammation [61]. Moreover, intestinal inflammation in the early stages of HFD may lead to obesity and related insulin resistance [59]. The high-dose RBPs intervention in this study reduced the abundance of *Campilobacterota* and *Deferribacteres*, indicating that RBPs have the potential to reduce intestinal inflammation and exert anti-obesity effects.

At the genus level, the high-dose RBPs intervention notably increased the abundance of *norank_f_Muribaculaceae*, *Muribaculum*, and *Prevotellaceas_UCG_001* in the feces of HFD-fed mice and increased the similarity to normal mice. The *Muribaculum* genus has been observed to be decreased in HFD mice and is related to body weight regulation [62]. *Norank_f_Muribaculaceae* is positively correlated with a reduction in liver steatosis and is an important SCFA-producing bacterium [63]. The abundance of *Prevotellaceas_UCG_001* was negatively correlated with GLU, TC, and HDL [64]. The bacteria of *Prevotellaceas_UCG_001* is proposed to affect the glucose and lipid metabolism of the host by producing secondary metabolites such as SCFAs [64]. Moreover, RBPs were shown to increase the abundance of *Alistipes*, *Odoribacter*, and *ASF356*. In another study, *Odoribacter* was positively correlated with the intestinal production of acetic acid in mice, and *Alistipes* and *ASF356* were positively correlated with IL-10 (an anti-inflammatory cytokine) [65]. Those findings reaffirmed the effect of RBPs on the intestinal inflammation at the phylum level in mice. Collectively, RBPs may ameliorate obesity and hyperlipidemia by improving the disorder of the gut microbiota in HFD-fed mice.

## 5. Conclusions

In the present study, we have provided solid evidence for the ameliorative effect of RBPs on obesity and hyperlipidemia in HFD-fed mice. Specifically, the RBPs intervention reduced body weight, the weight of adipose tissue, serum lipids, and hepatic fat accumulation in HFD-fed mice. Mechanistic studies suggested that RBPs exert anti-obesity and hypolipidemic activities by promoting lipid metabolism and resolving disorders in the gut microbiota. Specifically, RBPs could up-regulate the expression of p-AMPK, PPARα, and HSL, and down-regulate that of PPARγ in the liver to promote lipid catabolism. In addition, RBPs increased the abundance of gut microbiota, increased the abundance of microflora related to the synthesis of SCFAs (including *Bacteroides*, *Prevotellaceas_UCG_001*, *norank_f_Muribaculaceae*, and *Odoribacter*) and controlled the microflora related to intestinal inflammation (including reducing the abundance of *Deferribacteres* and increasing the abundance of *Alistipes* and *ASF356*) to exert anti-obesity and lipid-lowering activities. The present results demonstrated the potential to utilize RBPs as a functional component in the administration of obesity and hyperlipidemia. In addition, future studies are necessary to deeply explore the potential mechanisms, such as the transcriptional activities of PPARα and PPARγ, protein or mRNA expressions of lipid metabolism-related genes, inflammation, and production of intestinal microbial metabolites.

## Figures and Tables

**Figure 1 nutrients-14-05066-f001:**
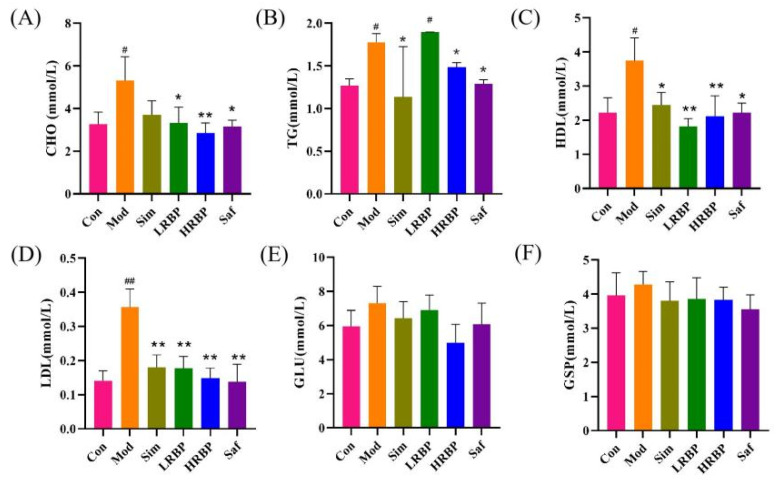
Effect of RBPs on blood lipid and blood sugar in Mice. (**A**–**D**) Serum levels of CHO, TG, HDL, and LDL. (**E**,**F**) Serum levels of GLU and GSP. ^#^
*p* < 0.05, ^##^
*p* < 0.01 compared to Con group, * *p* < 0.05, ** *p* < 0.01 compared to Mod group. Con, control group fed with normal diet; Mod, model group fed with HFD; Sim, positive control group fed with HFD and treated with simvastatin; LRBP, low dose RBPs group fed with HFD and treated with 200 mg/kg RBPs; HRBP, high dose RBPs group fed with HFD and treated with 600 mg/kg RBPs; Saf, safety evaluation group fed with normal diet and treated with 600 mg/kg RBPs.

**Figure 2 nutrients-14-05066-f002:**
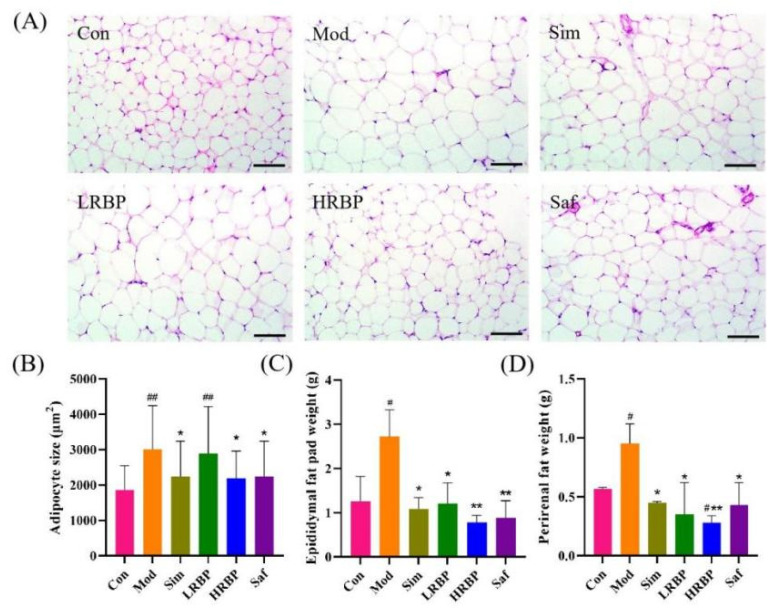
Effect of RBPs on adipose tissue in HFD-fed mice. (**A**) The results of HE staining of epididymal fat (scale bar = 100 µm). (**B**) Epididymal adipocyte size of mice. The result was measured with Image-J (Available online: https://imagej.net/Fiji/Downloads (accessed on 12 October 2022), USA National Institutes of Health, Bethesda, ML, USA). (**C**,**D**) Weights of epididymal and perirenal adipose tissue. ^#^
*p* < 0.05, ^##^
*p* < 0.01 compared to Con group, * *p* < 0.05, ** *p* < 0.01 compared to Mod group. Con, control group fed with normal diet; Mod, model group fed with HFD; Sim, positive control group fed with HFD and treated with simvastatin; LRBP, low dose RBPs group fed with HFD and treated with 200 mg/kg RBPs; HRBP, high dose RBPs group fed with HFD and treated with 600 mg/kg RBPs; Saf, safety evaluation group fed with normal diet and treated with 600 mg/kg RBPs.

**Figure 3 nutrients-14-05066-f003:**
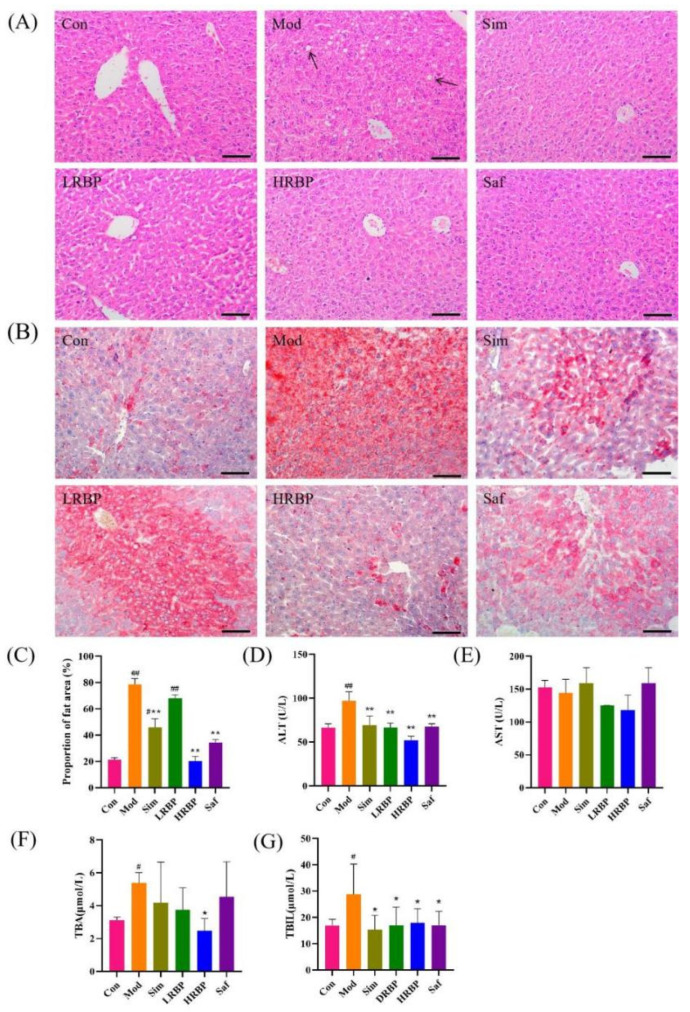
Protective effect of RBPs on the liver in mice. (**A**) The results of HE staining (scale bar = 100 µm). (**B**) The results of oil red O staining (scale bar = 100 µm). (**C**) The ratio of red area to the total area in oil red O staining results. The results were measured by Image-Pro Plus 6.0 software. (**D**–**G**) The content of ALT, AST, TBA, and TBIL in the serum of mice. ^#^
*p* < 0.05, ^##^
*p* < 0.01 compared to Con group, * *p* < 0.05, ** *p* < 0.01 compared to Mod group. Con, control group fed with normal diet; Mod, model group fed with HFD; Sim, positive control group fed with HFD and treated with simvastatin; LRBP, low dose RBPs group fed with HFD and treated with 200 mg/kg RBPs; HRBP, high dose RBPs group fed with HFD and treated with 600 mg/kg RBPs; Saf, safety evaluation group fed with normal diet and treated with 600 mg/kg RBPs.

**Figure 4 nutrients-14-05066-f004:**
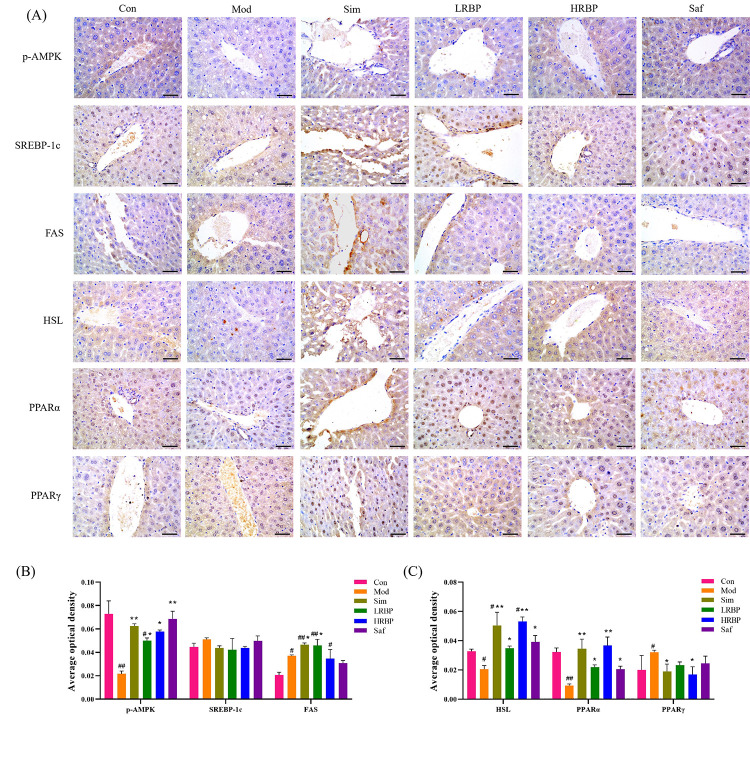
Effect of RBPs on the expression of lipid metabolism-related genes in the liver. (**A**) Immunohistochemical results of mice liver (scale bar = 50 µm). (**B**,**C**) Determination results of average optical density. The results were measured by Image-Pro Plus 6.0 software. ^#^
*p* < 0.05, ^##^
*p* < 0.01 compared to Con group, * *p* < 0.05, ** *p* < 0.01 compared to Mod group. Con, control group fed with normal diet; Mod, model group fed with HFD; Sim, positive control group fed with HFD and treated with simvastatin; LRBP, low dose RBPs group fed with HFD and treated with 200 mg/kg RBPs; HRBP, high dose RBPs group fed with HFD and treated with 600 mg/kg RBPs; Saf, safety evaluation group fed with normal diet and treated with 600 mg/kg RBPs.

**Figure 5 nutrients-14-05066-f005:**
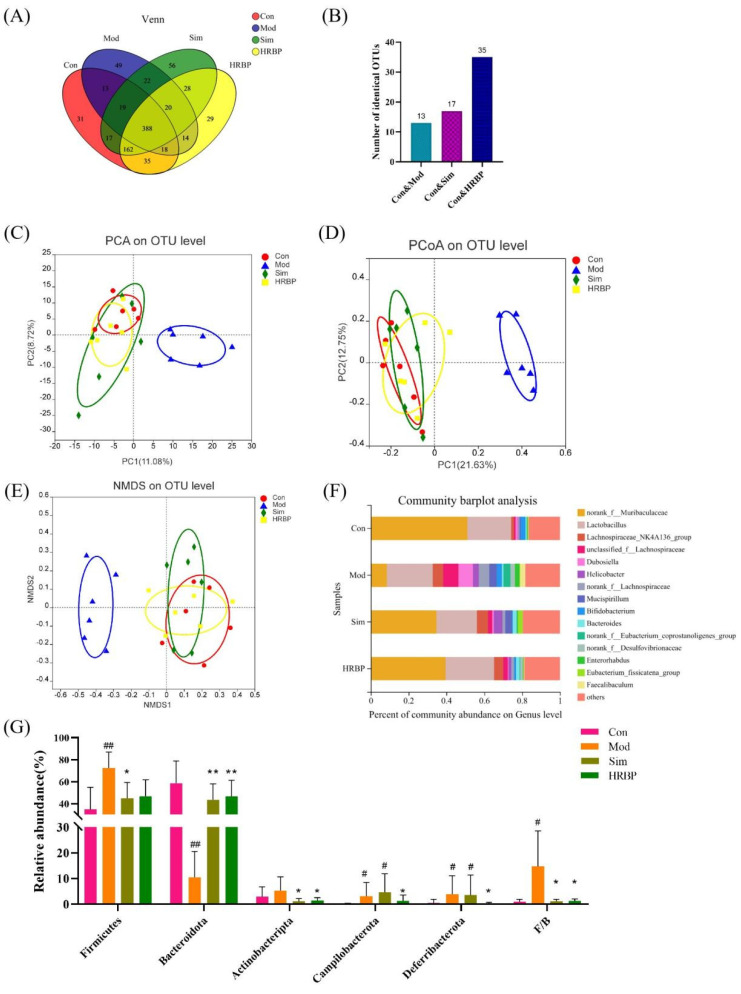
Effect of RBPs on the gut microbiota composition in hyperlipidemic mice. (**A**) Venn diagram. (**B**) Number of OTUs shared by each group and the Con group independently. (**C**–**E**) The results of beta diversity analysis. (**F**) Composition of the gut microflora at the genus level. (**G**) Relative abundance of the microbiota and F/B ratio at the phylum level. ^#^
*p* < 0.05, ^##^
*p* < 0.01 compared to Con group, * *p* < 0.05, ** *p* < 0.01 compared to Mod group. Con, control group fed with normal diet; Mod, model group fed with HFD; Sim, positive control group fed with HFD and treated with simvastatin; HRBP, high dose RBPs group fed with HFD and treated with 600 mg/kg RBPs.

**Figure 6 nutrients-14-05066-f006:**
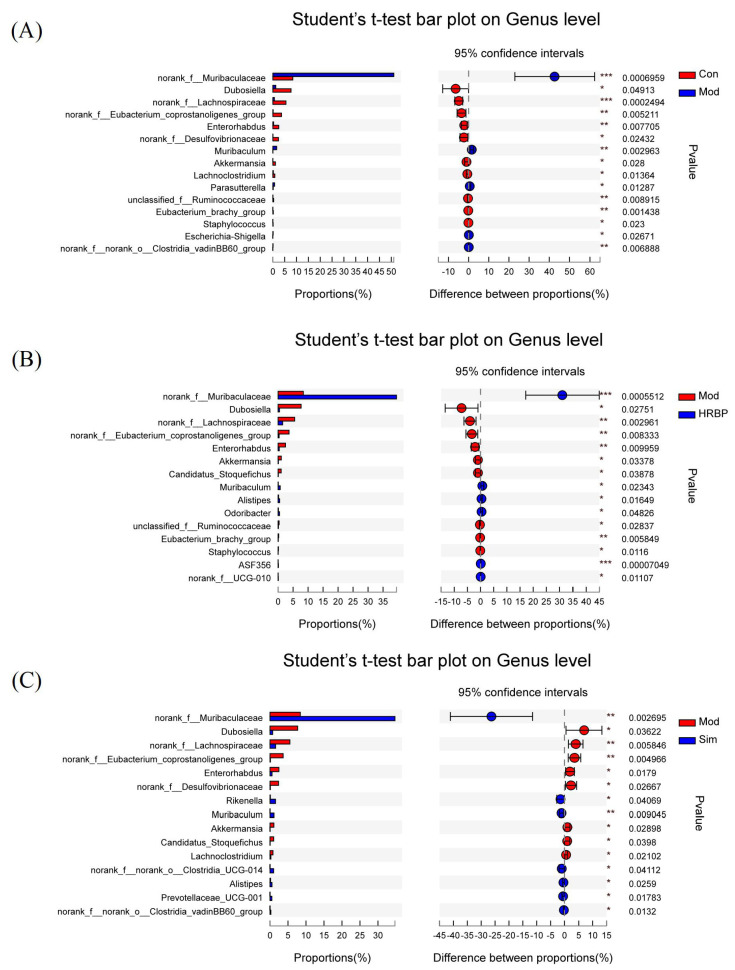
Student’s *t*-test bar plot of the genus proportions in different groups. (**A**–**C**) Student’s *t*-test bar plot of the genus proportions between the Con/Mod group, the Mod/HRBP group, and the Sim/Mod group. * *p* < 0.05, ** *p* < 0.01, *** *p* < 0.001 compared to Mod group. Con, control group fed with normal diet; Mod, model group fed with HFD; Sim, positive control group fed with HFD and treated with simvastatin; HRBP, high dose RBPs group fed with HFD and treated with 600 mg/kg RBPs.

**Table 1 nutrients-14-05066-t001:** Body weight and organ indices of the mice in each experimental group.

	Con	Mod	Sim	LRBP	HRBP	Saf
Weight before intervention (g)	48.30 ± 3.09	53.91 ± 2.10 ^#^	53.18 ± 2.63 ^#^	54.88 ± 3.55 ^#^	53.05 ± 5.52 ^#^	50.83 ± 5.04 *
Final body weight (g)	53.15 ± 0.91	61.20 ± 2.26 ^##^	57.35 ± 2.08 *	59.52 ± 4.84 ^#^	55.55 ± 3.75 *	54.14 ± 2.81 *
Cardiac index	0.55 ± 0.10	0.78 ± 0.04 ^#^	0.62 ± 0.06	0.75 ± 0.11	0.53 ± 0.06 *	0.58 ± 0.04 *
Liver index	4.14 ± 0.22	5.74 ± 0.55 ^#^	4.13 ± 0.17 *	3.97 ± 0.32 *	3.83 ± 0.29 *	4.44 ± 0.57
Spleen index	0.29 ± 0.07	0.26 ± 0.03	0.26 ± 0.07	0.31 ± 0.05	0.25 ± 0.04	0.23 ± 0.07
Renal index	1.44 ± 0.16	1.46 ± 0.14	1.35 ± 0.15	1.64 ± 0.15	1.46 ± 0.11	1.62 ± 0.23
Lung index	0.53 ± 0.08	0.51 ± 0.05	0.59 ± 0.02	0.52 ± 0.05	0.49 ± 0.01	0.42 ± 0.03

^#^*p* < 0.05, ^##^
*p* < 0.01 compared to Con group, * *p* < 0.05 compared to Mod group. Con, control group fed with normal diet; Mod, model group fed with HFD; Sim, positive control group fed with HFD and treated with simvastatin; LRBP, low dose RBPs group fed with HFD and treated with 200 mg/kg RBPs; HRBP, high dose RBPs group fed with HFD and treated with 600 mg/kg RBPs; Saf, safety evaluation group fed with normal diet and treated with 600 mg/kg RBPs.

**Table 2 nutrients-14-05066-t002:** Alpha diversity indices.

Estimators\Group	Con	Mod	Sim	HRBP
Coverage	0.9977	0.9983	0.9980	0.9976
Ace	462.03 ± 76.23	362.78 ± 43.25 ^#^	437.45 ± 72.43	478.32 ± 33.99 **
Chao	462.68 ± 79.42	362.69 ± 43.27 ^#^	437.96 ± 76.67	493.40 ± 37.42 **
Shannon	3.69 ± 0.53	3.51 ± 0.48	3.79 ± 0.53	3.96 ± 0.52
Simpson	0.0720 ± 0.0440	0.0861 ± 0.0567	0.0589 ± 0.0308	0.0616 ± 0.0651

^#^*p* < 0.05 compared to Con group, ** *p* < 0.01 compared to Mod group. Con, control group fed with normal diet; Mod, model group fed with HFD; Sim, positive control group fed with HFD and treated with simvastatin; HRBP, high dose RBPs group fed with HFD and treated with 600 mg/kg RBPs.

## Data Availability

The data presented in this study are available in this article and Appendix A.

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
