# Peer review of "The Ameliorative Effect and Mechanisms of Ruditapes philippinarum Bioactive Peptides on Obesity and Hyperlipidemia Induced by a High-Fat Diet in Mice"

_nutrients, 2022, doi:10.3390/nu14235066_

Round 1

Reviewer 1 Report

The authors conducted a study on the ameliorative effect and mechanisms of Ruditapes philip pinarum bioactive peptides on obesity and hyperlipidemia induced by a high-fat diet in mice. It appears that low and high concentrations of Ruditapes Philip pinarum were treated in a normal diet and a high-fat diet. However, since there are several groups, it is somewhat difficult to visualize the data in graphing. Add to the graph or specify in the figure legend.

Author Response

Reviewer 1:

The authors conducted a study on the ameliorative effect and mechanisms of Ruditapes philip pinarum bioactive peptides on obesity and hyperlipidemia induced by a high-fat diet in mice. It appears that low and high concentrations of Ruditapes Philip pinarum were treated in a normal diet and a high-fat diet. However, since there are several groups, it is somewhat difficult to visualize the data in graphing. Add to the graph or specify in the figure legend.

Answer: Thanks for your advice and sorry for the unclear labeled figures. As you have mentioned, the current study aimed to explore the improvement effect and mechanisms of RBPs on obesity and hyperlipidemia in mice caused by high-fat diet. We have six groups in this study: control group (Con, fed with normal diet and gavaged with normal saline), safety evaluation group (Saf, fed with normal diet and gavaged with 600 mg/kg RBPs), model group (Mod, fed with high-fat diet (HFD) and gavaged with normal saline), low-dose RBPs-treated group (LRBP, fed with HFD and gavaged with 200 mg/kg RBPs), high-dose RBPs-treated group (HRBP, fed with HFD and gavaged with 600 mg/kg RBPs) and simvastatin-treated positive control group (Sim, fed with HFD and gavaged with 2.6 mg/kg simvastatin). Based on your suggestion, we have reorganized the “2.2 Animal study” section, marked in red color. We also added corresponding legend under the figures and also re-marked the significance with “*” and “#” to make them more clear. As shown in line 148-152, line 244-247 and all the figure legends: “#p < 0.05, ##p < 0.01 compared to Con group, *p < 0.05, ** p < 0.01 compared to Mod group. Con, control group fed with normal diet; Mod, model group fed with HFD; Sim, positive control group fed with HFD and treated with simvastatin; LRBP, low dose RBPs group fed with HFD and treated with 200 mg/kg RBPs; HRBP, high dose RBPs group fed with HFD and treated with 600 mg/kg RBPs; Saf, safety evaluation group fed with normal diet and treated with 600 mg/kg RBPs.”

Reviewer 2 Report

1. Dosing of RBP given in mice and mentioned in line number 98 seems erroneous. It is 200, 600, 600 mg/kg BW. 

2. Data were analysed by ANOVA and not used post hoc t test. It's not sufficient.

3. Comparison between normal diet vs HFD has to be done and mentioned in graph.

4. Explanation of animal grouping is a bit confusing. It has to be written in meaningful and comprehensive way.

Author Response

Reviewer 2:

1.Dosing of RBP given in mice and mentioned in line number 98 seems erroneous. It is 200, 600, 600 mg/kg BW. 

Answer: Thanks for your helpful advice. We confirmed the treatment dose was correct. While, our description was unclear to make it confusing. Based on your suggestion, we have made corresponding modification on the “Animal study” section, as shown in line 91-99: “After one week of adaptation, the mice were randomly divided into two groups that fed with normal diet (n = 16) and 40% high-fat diet (HFD, the composition was shown in Supplemental Table S2) (n = 32), respectively. Nine weeks later, mice fed with normal diet were randomly divided into two groups (n = 8 each): Control (Con) group, gavaged with normal saline; safety evaluation (Saf) group, gavaged with 600 mg/kg RBPs. The mice fed with HFD were randomly divided into four groups (n = 8 each): Model (Mod) group, gavaged with normal saline; Low dose RBPs (LRBP) group, gavaged with 200 mg/kg RBPs; High dose RBPs (HRBP) group, gavaged with 600 mg/kg RBPs; Simvastatin (Sim) group, gavaged with 2.6 mg/kg simvastatin.”

2.Data were analysed by ANOVA and not used post hoc t test. It's not sufficient.

Answer: Thanks for your advice. It’s a typo. We used Tukey’s post hoc test, while we failed to describe in the method. As suggested, we have added the description in the “Statistical analysis” section, as shown in line 132-134: “Unless otherwise specified, all data were analyzed by one-way analysis of variance (ANOVA) followed by Tukey’s post hoc test.”

3.Comparison between normal diet vs HFD has to be done and mentioned in graph.

Answer: We have compared all the experimental groups. However, our label of significance might be unclear. Based on your suggestion, we have re-marked the significance with “*” and “#” to make them more clear.

4.Explanation of animal grouping is a bit confusing. It has to be written in meaningful and comprehensive way.

Answer: Thanks for your helpful advice. We have modified the description of animal grouping in the “Materials and Methods section based on your kind suggestion, as shown in line 91-101: “After one week of adaptation, the mice were randomly divided into two groups that fed with normal diet (n = 16) and 40% high-fat diet (HFD, the composition was shown in Supplemental Table S2) (n = 32), respectively. Nine weeks later, mice fed with normal diet were randomly divided into two groups (n = 8 each): Control (Con) group, gavaged with normal saline; safety evaluation (Saf) group, gavaged with 600 mg/kg RBPs. The mice fed with HFD were randomly divided into four groups (n = 8 each): Model (Mod) group, gavaged with normal saline; Low dose RBPs (LRBP) group, gavaged with 200 mg/kg RBPs; High dose RBPs (HRBP) group, gavaged with 600 mg/kg RBPs; Simvastatin (Sim) group, gavaged with 2.6 mg/kg simvastatin. The mice received corresponding gavage every day at fixed time for 6 weeks and the treatment dose was referenced to previous publications [20, 26].”